# Convergent Akt activation drives acquired EGFR inhibitor resistance in lung cancer

Kirstine Jacobsen[1], Jordi Bertran-Alamillo[2], Miguel Angel Molina[2], Cristina Teixidó[2], Niki Karachaliou[3], Martin Haar Pedersen[1], Josep Castellví[2], Mónica Garzón[2], Carles Codony-Servat[2], Jordi Codony-Servat[2], Ana Giménez-Capitán[2], Ana Drozdowskyj[4], Santiago Viteri[5], Martin R. Larsen[6], Ulrik Lassen[7], Enriqueta Felip[8], Trever G. Bivona[9,10], Henrik J. Ditzel [1,11] & Rafael Rosell[2,5,12,13]

Non-small-cell lung cancer patients with activating epidermal growth factor receptor (EGFR) mutations typically benefit from EGFR tyrosine kinase inhibitor treatment. However, virtually all patients succumb to acquired EGFR tyrosine kinase inhibitor resistance that occurs via diverse mechanisms. The diversity and unpredictability of EGFR tyrosine kinase inhibitor resistance mechanisms presents a challenge for developing new treatments to overcome EGFR tyrosine kinase inhibitor resistance. Here, we show that Akt activation is a convergent feature of acquired EGFR tyrosine kinase inhibitor resistance, across a spectrum of diverse, established upstream resistance mechanisms. Combined treatment with an EGFR tyrosine kinase inhibitor and Akt inhibitor causes apoptosis and synergistic growth inhibition in multiple EGFR tyrosine kinase inhibitor-resistant non-small-cell lung cancer models. Moreover, phospho-Akt levels are increased in most clinical specimens obtained from EGFR-mutant non-small-cell lung cancer patients with acquired EGFR tyrosine kinase inhibitor resistance. Our findings provide a rationale for clinical trials testing Akt and EGFR inhibitor co-treatment in patients with elevated phospho-Akt levels to therapeutically combat the heterogeneity of EGFR tyrosine kinase inhibitor resistance mechanisms.

[1] Department of Cancer and Inflammation Research, Institute of Molecular Medicine, University of Southern Denmark, 5000 Odense, Denmark. [2] Laboratory of Oncology, Pangaea Biotech, Quiron Dexeus University Hospital, 08028 Barcelona, Spain. [3] Instituto Oncológico Dr. Rosell, University Hospital Sagrat Cor, 08029 Barcelona, Spain. [4] Pivotal, 28023 Madrid, Spain. [5] Instituto Oncológico Dr. Rosell, Quiron-Dexeus University Hospital, 08028 Barcelona, Spain. [6] Protein Research Group, Institute of Biochemistry and Molecular Biology, University of Southern Denmark, 5230 Odense, Denmark. [7] Phase I Unit, Rigshospitalet, 2100 Copenhagen, Denmark. [8] Department of Medical Oncology, Vall D´Hebron, 08035 Barcelona, Spain. [9] Department of Medicine, Division of Hematology and Oncology, University of California, San Francisco, CA 94158, USA. [10] Helen Diller Comprehensive Cancer Center, University of California, San Francisco, CA 94158, USA. [11] Department of Oncology, Odense University Hospital, 5000 Odense, Denmark. [12] Catalan Institute of Oncology, Hospital Germans Trias i Pujol, 08916 Badalona, Spain. [13] Germans Trias i Pujol, Health Sciences Institute and Hospital, Campus Can Ruti, 08916 Badalona, Spain. Correspondence and requests for materials should be addressed to T.G.B. (email: trever.bivona@ucsf.edu) or to H.J.D. (email: hditzel@health.sdu.dk)

Lung cancer is the leading cause of cancer mortality worldwide[1]. Mutations in epidermal growth factor receptor (EGFR), most commonly deletions in exon 19 (delE746-750) or substitution of arginine for leucine (L858R) in exon 21, are present in ~17% of tumors in patients with pulmonary adenocarcinoma[2] and confer sensitivity to the EGFR-tyrosine kinase inhibitors (TKIs) gefitinib[3, 4], erlotinib[5, 6] or afatinib[7, 8]. The principal downstream pathways mediating the oncogenic effects of EGFR are extracellular signal–regulated kinase 1 and 2 (ERK1/2) via Ras, Akt via phosphatidylinositol 3-kinase (PI3K), and

signal transducer and activator of transcription 3 (STAT3) via Janus kinase 2 (JAK2)[9].

Acquired resistance substantially limits the clinical efficacy of EGFR TKIs. Although ~70% of EGFR-mutant non-small-cell lung cancer (NSCLC) patients respond to first-line EGFR-TKI treatment, the majority of them do not achieve complete responses and virtually all patients develop acquired resistance and lethal disease progression[6]. A diversity of EGFR-TKI resistance mechanisms has been described, of which the most frequent mechanism of resistance to EGFR-TKI treatment is the

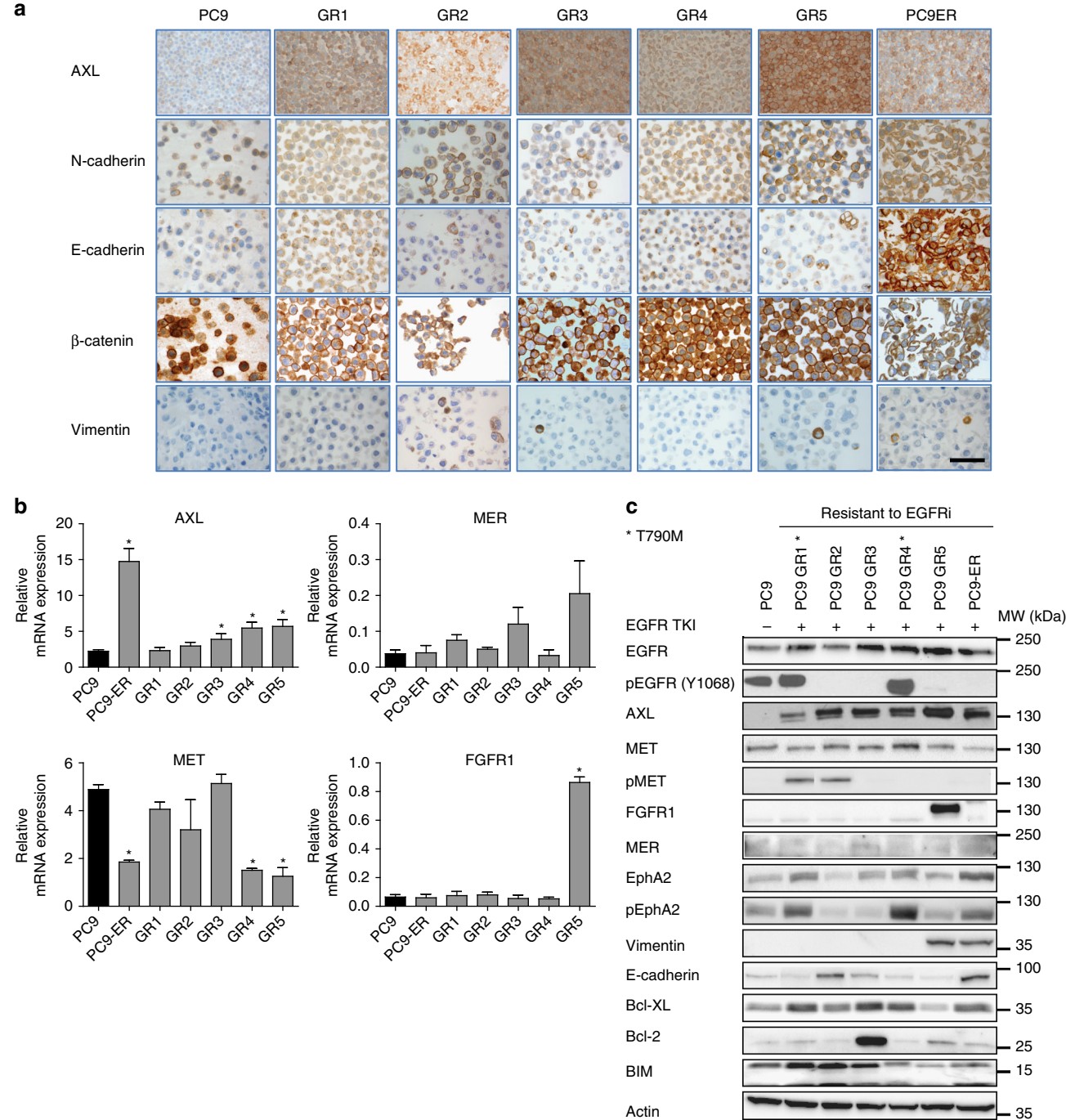

**Fig. 1** Expression of proteins related to acquired resistance in parental and resistant PC9 cell lines. **a** Immunohistochemistry of selected EMT markers. *Scale bar*, 100 μm. **b** Relative mRNA expression of selected genes. Experiments were performed in triplicates and results are shown as mean ± SD. *Asterisks* indicate significant difference from PC9 in a Student's t-test. **c** Western blot analysis of selected proteins. In all cases, PC9 cells were grown without drug, whereas PC9-GR1-5 were grown in 5 μM gefitinib and PC9-ER was grown in 30 μM erlotinib

secondary mutation in exon 20 of EGFR, T790M[10, 11]. Other mechanisms include amplification, overexpression, and autocrine loops involving MET proto-oncogene (MET), erb-b2 receptor tyrosine kinase 2 (ErbB2), ephrin type-A receptor 2 (EphA2), fibroblast growth factor receptor (FGFR) and the members of the TAM receptor tyrosine kinase (RTKs), Mer and AXL[12–15]. In addition, we have shown that activation of NF-κB rescues EGFR-mutant lung cancer cells from EGFR-TKI treatment[16]. Finally, BRAF and PIK3CA mutations, conversion to small-cell-lung cancer and occurrence of epithelial-to-mesenchymal transition (EMT) have also been associated with acquired resistance to EGFR-TKI in NSCLC[12]. Certain EGFR-mutant NSCLCs harbor multiple mechanisms of EGFR-TKI resistance[17, 18]. In these cases, the co-occurrence of multiple resistance mechanisms is likely to lessen the therapeutic impact of targeting each individual resistance-promoting alteration. Additionally, which specific resistance alteration(s) will arise and promote EGFR-TKI resistance in individual patients is currently largely unpredictable at the outset of therapy. Hence, the diversity and unpredictability of EGFR-TKI resistance mechanisms presents a major challenge for efficiently developing new treatment regimens that can overcome EGFR-TKI resistance in patients.

Activation of the Akt pathway is a common feature in human cancers and leads to increased cell survival, growth, and proliferation[19]. V-akt murine thymoma viral oncogene homologs 1, 2 and 3 (Akt1, Akt2, and Akt3) comprise the Akt family of serine-threonine kinases, which are tethered to the membrane via interaction with phosphatidylinositol-3,4,5-triphosphate (PIP$_3$) lipids[20], and activated by phosphorylation on threonine 308 (Thr308) by 3-phosphoinositide-dependent protein kinase 1 (PDK1)[21] and serine 473 (Ser473) by the mammalian target of rapamycin complex 2 (mTORC2)[22]. Activated Akt phosphorylates many downstream targets, including forkhead box O3 (FOXO3) and proline-rich Akt substrate of 40 kDa (PRAS40)[23–26]. Several small molecule drugs targeting components of the Akt pathway have been developed and are being tested in patients[27]. Interestingly, first-line sensitivity to EGFR TKIs in NSCLC has been associated with pre-existent Akt activation that is suppressed by EGFR inhibition, while treatment with EGFR TKIs failed to block Akt signaling in tumor cells intrinsically resistant to these drugs[28–31]. In addition, the combination of a PIK3-mTOR inhibitor with a MEK inhibitor has been reported to induce apoptosis in EGFR-TKI naïve EGFR-mutant NSCLC cell lines and xenografts, although the combination of an Akt and a MEK inhibitor failed to have this effect in this TKI-naive context[32]. Despite evidence suggesting a general role for PI3K-AKT-mTOR pathway signaling in EGFR-mutant NSCLC, whether Akt activation, specifically, can drive acquired EGFR-TKI resistance has not been clearly demonstrated. Furthermore, the hypothesis that Akt activation functions as a convergent, resistance-driving signaling event across a spectrum of EGFR-mutant NSCLCs that harbor otherwise diverse, established EGFR-TKI resistance-promoting mechanisms has not been tested.

Here, we show that Akt pathway activation is a convergent feature in EGFR-mutant NSCLCs with acquired resistance to EGFR TKIs that may be caused by diverse underlying mechanisms. This convergent resistance-promoting function of Akt activation occurred in the presence of one or more different resistance mechanisms such as amplification, overexpression, and activation of MET, EphA2, FGFR, Mer, and AXL or the presence of the T790M mutation. We show that combined treatment with Akt and EGFR inhibitors in resistant EGFR-mutant NSCLC models synergistically inhibits growth in this heterogeneous molecular background. We also show that phospho-Akt (pAkt) is increased in the majority of EGFR-mutant

patients after progression on EGFR TKIs, and also that high levels of pAkt in patients prior to EGFR-TKI treatment correlates with poor initial therapeutic responses. Thus, convergent Akt activation is a previously unrecognized therapeutic target whose inhibition has the unique potential to combat the profound molecular heterogeneity underlying EGFR-TKI clinical resistance. Our findings provide a new rationale to test combined Akt plus EGFR inhibitor treatment, specifically in EGFR-mutant NSCLCs with high pAkt levels, to overcome acquired EGFR-TKI resistance and enhance patient survival.

## Results

**Establishment of PC9-derived cell lines resistant to EGFR TKIs.** Six EGFR-TKI-resistant cell lines were generated by treating EGFR-TKI-sensitive PC9 cells, which harbor the EGFR exon 19 deletion, with increasing concentrations of gefitinib (GR1-5) or erlotinib (ER). Sequencing analyses revealed that all six cell lines retained the EGFR exon 19 deletion (Supplementary Table 1), while the T790M mutation emerged in two (PC9-GR1 and PC9-GR4 at allelic fractions of 25 and 38% respectively). No mutations were detected in HER2, PIK3CA, BRAF of KRAS. We determined the sensitivity of the parental and resistant cell lines to a variety of EGFR TKIs (Supplementary Table 2). The half-maximal inhibitory concentration (IC50) for gefitinib and erlotinib of parental PC9 cells was in the nanomolar range compared to 4–29 μM in the resistant cell lines. In T790M-negative cells (PC9-ER, -GR2, -GR3, and -GR5), acquisition of resistance to EGFR-TKI also led to insensitivity to the second- (afatinib or dacomitinib) and third- (osimertinib (AZD9291)) generation EGFR TKIs. The T790M-positive cell lines (PC9-GR1 and -GR4) remained sensitive to osimertinib and became only moderately resistant to afatinib and dacomitinib. Acquisition of the resistant phenotype was often associated with morphological changes and altered anchorage-independent growth, invasiveness, and migration (Supplementary Fig. 1).

**Common resistance mechanisms in PC9-derived cell lines.** To delineate the underlying resistance mechanism of the EGFR-TKI-resistant cell lines, we initially investigated previously reported acquired resistance mechanisms other than T790M mutation using fluorescence in situ hybridization, immunohistochemistry (IHC), quantitative real-time polymerase chain reaction (qRT-PCR) and wsestern blotting. Regarding mRNA and protein expression, AXL upregulation was the most common molecular event, while FGFR1 overexpression and EphA2 and MET activation were also found in some of the resistant cell lines (Figs. 1a–c). In contrast, no significant differences were observed in the case of FGFR2, MER (Figs. 1b, c), ErbB2, or ErbB3 (Supplementary Fig. 2). Alterations in proteins related to apoptosis (Bcl-2, Bcl-xl, and BIM) and EMT (E-cadherin, vimentin, N-cadherin, and beta-catenin) were also observed (Figs. 1a, c). EGFR was equally amplified in the parental PC9 and all the resistant cell lines, while there was no MET, FGFR1, or ErbB2 amplification in any of them. In all six resistant cell lines, two or more possible mechanisms of acquired resistance were simultaneously present (Supplementary Table 3), but each cell line had a unique profile in terms of the specific repertoire of established resistance mechanisms.

**Quantitative proteomic analysis of PC9-ER resistant cells.** To further identify mechanisms associated with EGFR-TKI resistance, we used quantitative proteomics to compare the proteomes of parental and PC9-ER cells. To increase the number of proteins identified, samples were first separated into soluble and membrane-associated proteins and subsequently fractionated

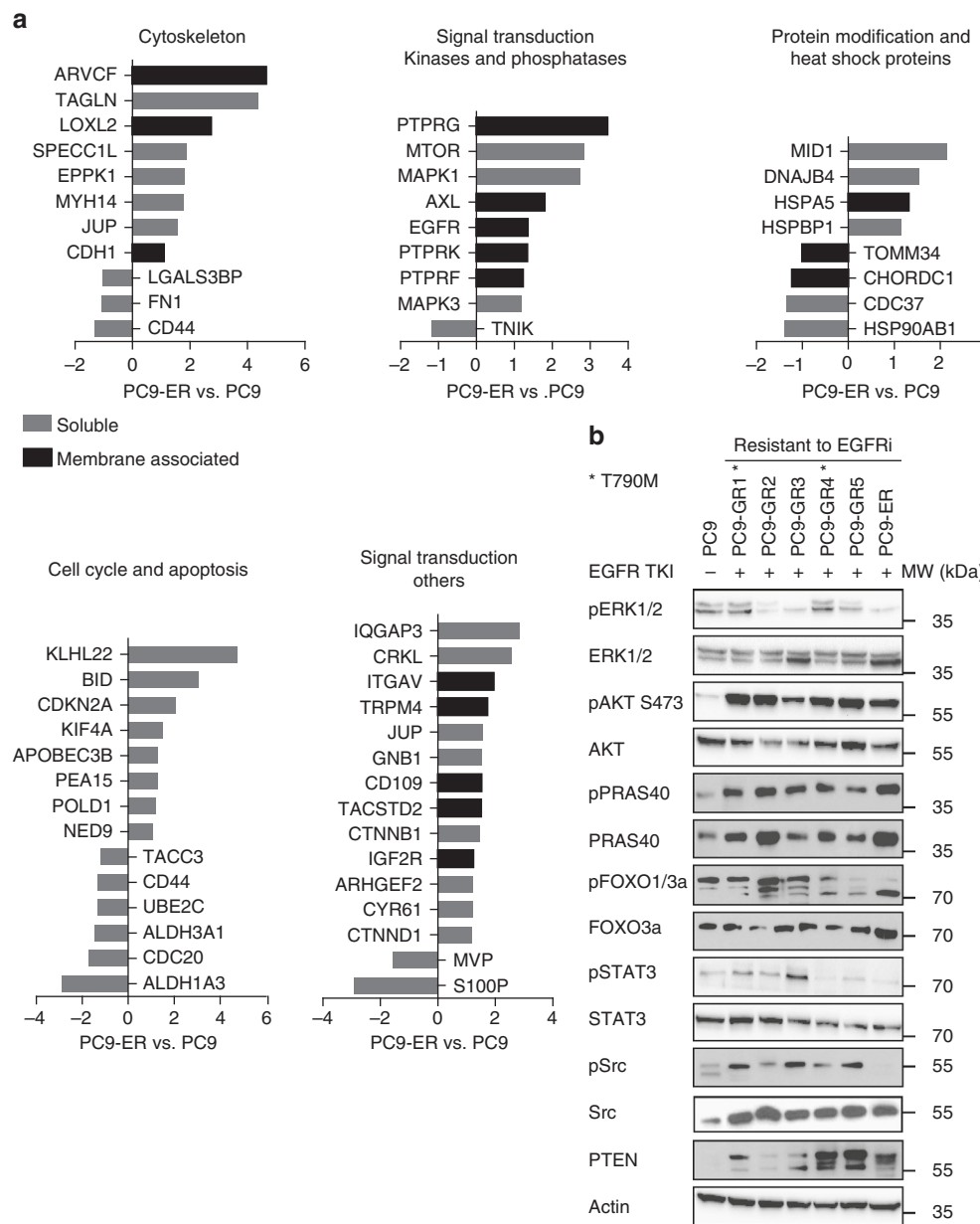

**Fig. 2** Proteomics and western blot analyses of signal transduction proteins in the parental and EGFR-TKI-resistant PC9 cells. **a** Distribution of ratios between differentially expressed proteins of the Akt pathway in PC9-ER compared to PC9 in the membrane-associated and soluble fractions derived from the mass spectrometric analysis (data are log2 transformed). Ingenuity Pathway Analysis software (Qiagen) was used for classifying the proteins in different functional groups. **b** Western blot analysis of key signal transduction proteins in the parental PC9 cells growing in absence of drugs and the six EGFR-TKI-resistant PC9-derived cell lines cultured with EGFR TKI

by hydrophilic interaction liquid chromatography (HILIC). Each of the ten HILIC fractions was subsequently analyzed by liquid chromatography tandem mass spectrometry (LC–MS/MS). PC9 and PC9-ER cells were analyzed in biological triplicates. A total of 3535 proteins were identified based on two unique peptides (FDR 1%) with an overlap of 63.5% between the membrane-associated and soluble fractions. Based on the distribution of the log2 transformed ratios of differentially expressed proteins, 2-fold was chosen as the differential expression threshold. A total of 357 proteins exhibited altered expression, of which 247 were upregulated and 110 downregulated, in the EGFR-TKI-resistant PC9-ER cells compared to the EGFR-TKI-sensitive, parental PC9 cells.

Next, the differentially expressed proteins between EGFR-TKI-sensitive and -resistant cells were classified according to functional subgroups (Supplementary Fig. 3a–c). The extensive number of proteins differentially expressed and the variety of cell processes affected revealed an entire reprogramming of the cell machinery associated with the emergence of resistance. Four membrane receptors were upregulated in the resistant cell lines: AXL, integrin alpha-V (ITGAV) and, to a lesser extent, insulin-like growth factor 2 receptor and EGFR. The result for AXL was coincident with the upregulation shown by western blotting, IHC and qRT-PCR. Regarding intracellular signal transduction, the most relevant finding was the 7-fold upregulation of the mammalian target of rapamycin (mTOR). We also observed a significant overexpression of key proteins such as v-crk avian sarcoma virus CT10 oncogene homolog-like (CRKL), ERK1, ERK2, several G-proteins and GTPase activators, some phosphatases and the ubiquitin E3 ligase Midline 1 (MID1; Fig. 2a).

The roles of mTOR, ERK1 and ERK2 are widely known, while CRKL is an oncogene that constitutes a major convergence point in tyrosine kinase signaling. In addition, CRKL amplification has been described as a mechanism of acquired resistance to EGFR TKIs in some EGFR-mutant NSCLCs, acting via activation of the ERK and Akt pathways[33]. MID1 is known to activate the Akt/

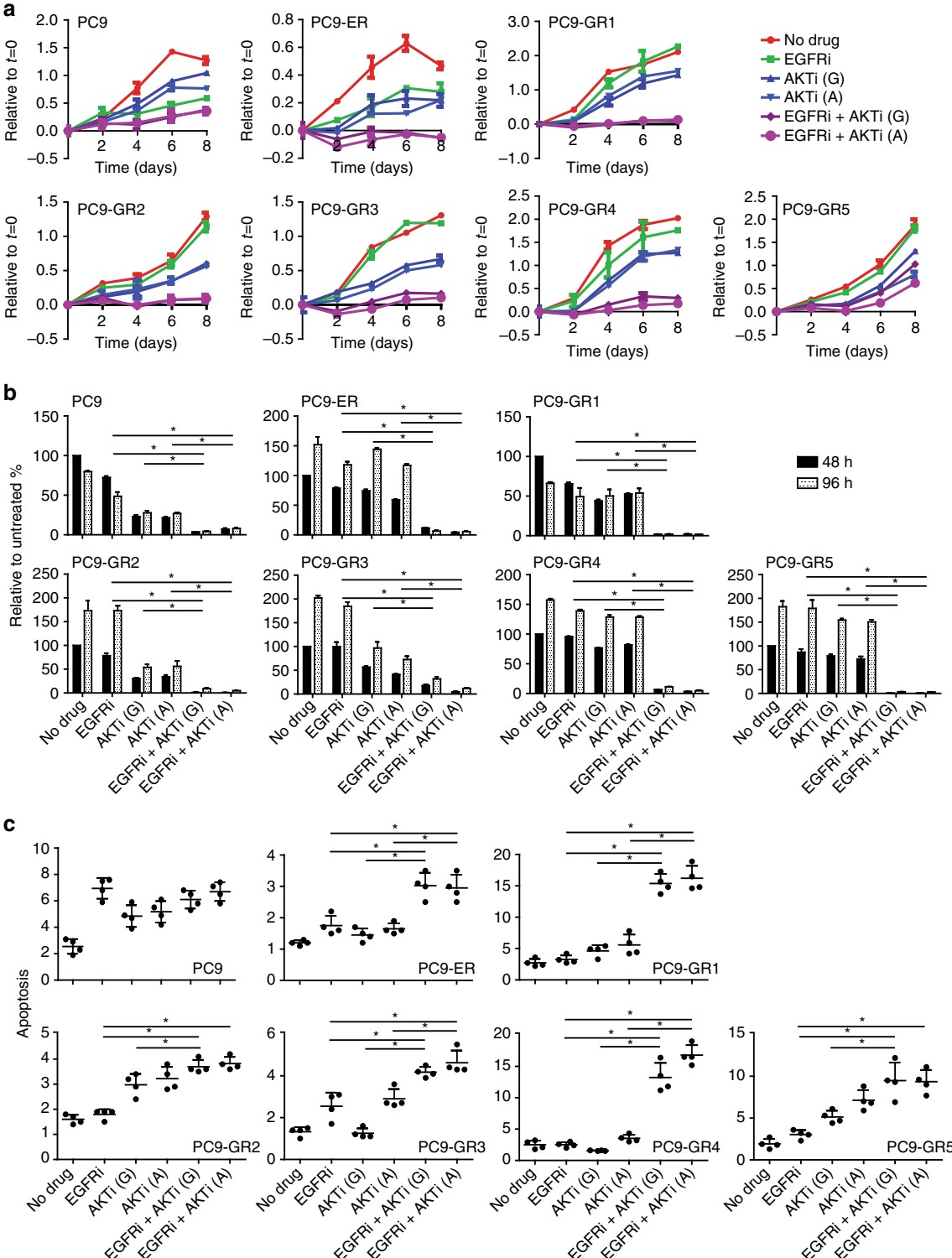

**Fig. 3** Synergistic effects of EGFR TKIs combined with Akt inhibitors in parental and EGFR-TKI-resistant PC9 cells. The combined effects of EGFR inhibitors (erlotinib for PC9-ER and gefitinib for parental PC9 and PC9-GR1-5) and Akt inhibitors (GSK2141795 or AZD5363 for all cell lines) were assessed by **a** crystal violet viability assay performed over 8 days, **b** BrdU incorporation assay performed after 48 and 96 h incubation, and **c** apoptosis assay performed after 96 h incubation. Concentrations of drugs were chosen to be sub-inhibitory to explore the synergistic potential: erlotinib 30 μM; gefitinib 5 μM (except for PC9 where 40 nM was used); GSK2141795 (G) 2.5 μM; AZD5363 (A) 35 μM, (except for GR2 and GR5 where 1.25 and 3 μM were employed). All experiments were conducted in quadruplicates (**a**), or triplicates (**b**, **c**), and results are shown as mean ± SD. *Asterisks* indicate significant difference in ANOVA one-way test ($p < 0.05$) for the drug combination-treated cells compared to cells treated with either drug alone at the same time point

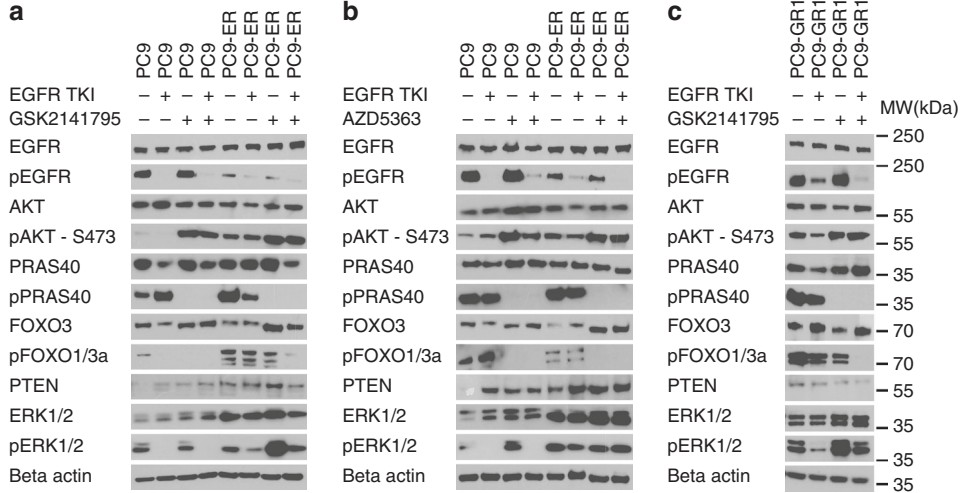

**Fig. 4** Effect of EGFR TKIs and Akt inhibitors on key signal transduction proteins in parental and resistant PC9 cells. **a** Western blot analysis of PC9 and PC9-ER treated for 4 h with erlotinib (30 µM), GSK2141795 (2.5 µM) or the combination; **b** Western blot analysis of PC9 and PC9-ER treated for 4 h with erlotinib (30 µM), AZD5363 (35 µM), or the combination; **c** Western blot analysis of PC9-GR1 treated for 4 h with gefitinib (5 µM), GSK2141795 (2.5 µM), or the combination

mTOR pathway through degradation of the catalytic subunit of the tumor suppressor phosphatase 2A (PP2A)[34].

We subsequently performed a global analysis of the selected proteins using the Ingenuity Pathway Analysis software. Regarding signal transduction, the most relevant finding was the upregulation of the PI3K-Akt-mTOR signaling pathway in the EGFR-TKI-resistant vs. sensitive cells. This finding was in agreement with the upregulation of several proteins involved in or activated by the PI3K-Akt-mTOR signaling pathway revealed by the individual analyses, such as the membrane receptors AXL, ITGAV, and IGFR2, the downstream signaling proteins CRKL and mTOR, the ubiquitin ligase MID1 or the lysine oxidase LOXL2 (Fig. 2a).

**Activation of the Akt pathway in PC9-derived resistant cell lines.** In view of the proteomics results, we explored the activation status of Akt signal transduction pathways in the parental and resistant cell lines (Fig. 2b) and also studied ERK and STAT3 activation. The levels of pAkt and its downstream effector phospho-PRAS40 (pPRAS40) were elevated in the presence of EGFR TKI in all the resistant cell lines, regardless of T790M, AXL, MET, EphA2, or FGFR1 status, while significant levels pFOXO1/3a were also present, indicating that Akt is a convergent hub for a variety of acquired resistance mechanisms. In contrast, key effector proteins of ERK and STAT3 signaling pathways did not show the same consistent behavior (Fig. 2b). The two resistant T790M-positive cell lines had increased levels of phospho-ERK (pERK) in the presence of EGFR-TKI, while ERK activation was strongly inhibited by the drug in T790M-negative cell lines. Finally, phosphorylated levels of STAT3 and c-Src were only elevated in one and three resistant cell lines, respectively.

To gain insight in the mechanisms leading to Akt activation, resistant cells were treated with selected targeted agents (Supplementary Fig. 4). The FGFR inhibitor nintedanib and the MET inhibitor crizotinib induced a dose-dependent decrease of pAkt levels in PC9-GR5 and PC9-GR2, respectively. PC9-GR5 cells overexpress FGFR1 and PC9-GR2 cells show MET activation (Fig. 1b). In contrast, the AXL inhibitor BGB324 had little or no effect on Akt phosphorylation when added to the

AXL-overexpressing PC9-ER cells. Finally, the T790M-positive PC9-GR1 cells, which also show MET activation, were treated with crizotinib and osimertinib. Both drugs induced a moderate, dose-dependent decrease in pAKT levels (Supplementary Fig. 4d).

**EGFR TKI and an Akt inhibitor elicits synergistic growth inhibition.** Of all the RTKs and signal transduction proteins investigated, only Akt and PRAS40 were consistently activated in the six resistant cell lines in the presence of EGFR TKI. Consequently, we assessed the effects of two different Akt inhibitors, GSK2141795 and AZD5363, on cell growth and signaling. When tested as single agents, both inhibited cell growth with IC50s of 5–15 µM (Supplementary Fig. 5a). When combined with an EGFR TKI, the Akt inhibitors elicited a synergistic growth inhibitory effect on the six resistant cell lines, which was significantly less strong in the parental PC9 (Fig. 3a, b and Supplementary Table 4). This synergistic effect was observed at all incubation times tested and with a variety of methods: including crystal violet, measuring the number of cells (Fig. 3a), CellTiterBlue, estimating the metabolic activity of living cells (Supplementary Fig. 5b) and BrdU incorporation, measuring cell proliferation (Fig. 3b). Finally, addition of an Akt inhibitor to the EGFR TKI also had a significant enhancing effect on apoptosis as measured by a DNA fragmentation assay (Fig. 3c). In view of these results, we examined the effects of the Akt inhibitors and EGFR-TKI treatment on signal transduction pathways in three cell lines: PC9, PC9-ER (T790M-negative) and PC9-GR1 (T790M-positive; Fig. 4a–c). EGFR TKIs alone suppressed phospho-EGFR (pEGFR), and pERK to a lesser extent, while having little or no effect on activation of Akt (S473) and its downstream effectors PRAS40 and FOXO1/3 A. Conversely, Akt inhibitors did not modify phosphorylation levels of EGFR or ERK, but significantly decreased Akt pathway activation. Both GSK2141795 and AZD5363 completely blocked PRAS40 phosphorylation, while only AZD5363 suppressed FOXO1/3A activation in the resistant cell lines. Both inhibitors caused increased Akt phosphorylation in all cell lines, as shown previously[35, 36], but the reduced phosphorylation levels of PRAS40 and FOXO1/3A clearly indicated inhibition of Akt

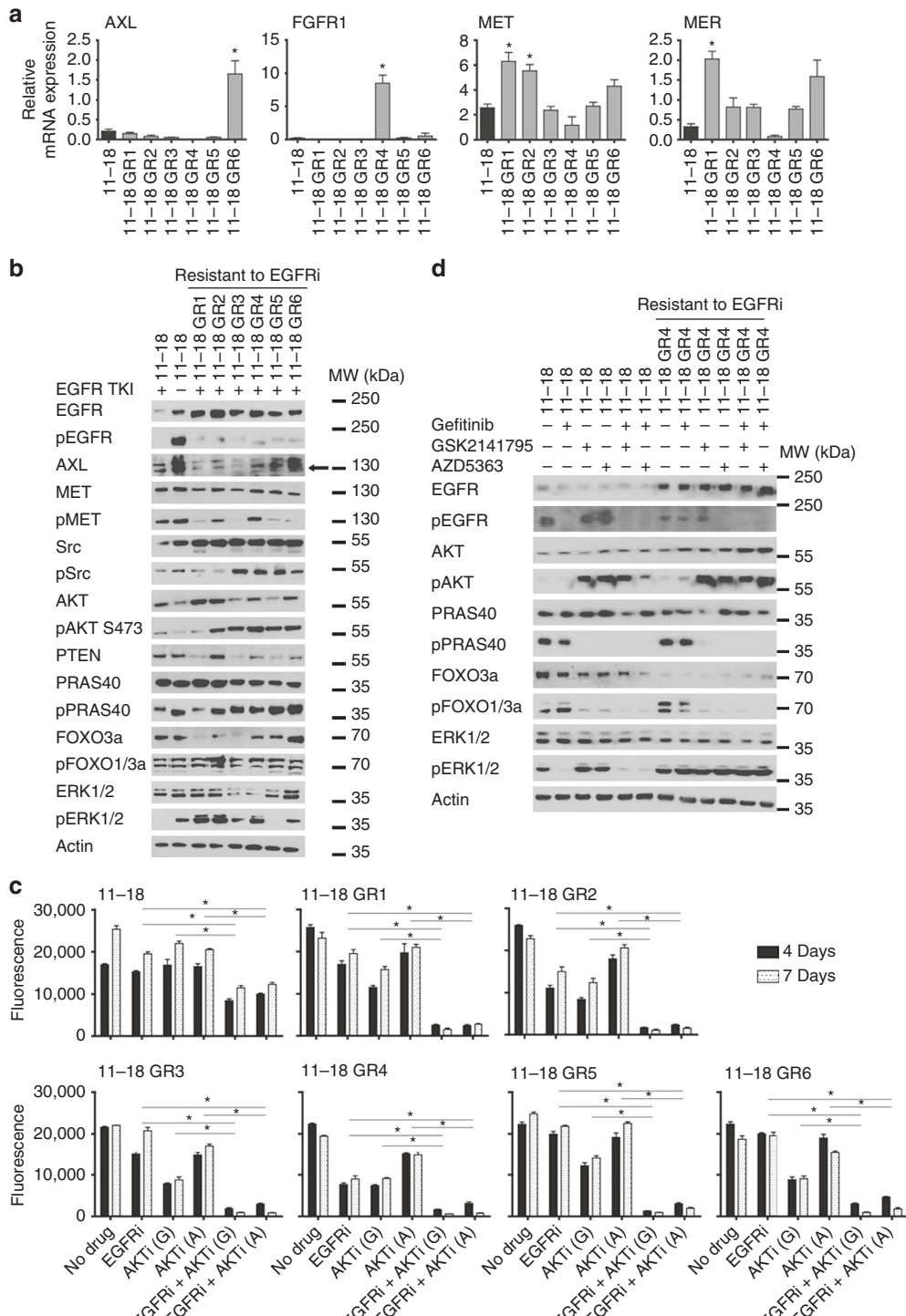

**Fig. 5** Expression of selected proteins in parental and resistant 11–18 cells and synergistic effects of EGFR TKIs and Akt inhibitors. **a** Relative mRNA expression of selected genes. Experiments were at least performed in triplicates and results are shown as mean ± SD. *Asterisks* indicate significant difference from parental PC9 in a Student's *t*-test ($p < 0.05$). **b** Western blot analysis of selected proteins after 4 h stimulation with 5 μM gefitinib. **c** Concentrations of drugs were chosen to be sub-inhibitory to explore the synergistic potential: gefitinib (EGFRi) 10 μM (except for 18-11 where 100 nM was used); GSK2141795 (G) 2.5 μM; AZD5363 (A) 35 μM. All experiments were conducted in seven replicates and results are shown as mean ± SD. *Asterisks* indicate significant difference in ANOVA one-way test ($p < 0.05$) for the drug combination-treated cells compared to cells treated with either drug alone at the same time point. **d** Western blot analysis of 11–18 and 11–18 GR4 treated for 4 h with gefitinib (10 μM), GSK2141795 (2.5 μM) or the combination

kinase activity. Combining an EGFR TKI with either GSK2141795 or AZD5363 resulted in simultaneous inhibition of EGFR, PRAS40 and FOXO1/3A phosphorylation (Fig. 4a–c). We also considered other targets of the Akt pathway and included

a PI3K-mTOR inhibitor, GSK2141458, to be combined with EGFR inhibitors (Supplementary Fig. 5c). However, less synergism was observed with the PI3K-mTOR inhibitor compared to the Akt inhibitor GSK2141795 in combination with an

EGFR inhibitor, indicating that Akt inhibition may be the most effective strategy for blocking the pathway and survival in these cells (Supplementary Table 5).

**Activation of Akt in 11–18-derived EGFR-TKI-resistant cell lines.** To validate our findings in an additional and independent system, we generated and characterized six other EGFR-TKI-resistant cell lines derived from the 11–18 NSCLC cell line, which harbors the other major EGFR-TKI-sensitizing EGFR mutation observed in NSCLC patients, EGFR L858R. The IC50s for gefitinib were 40–200-fold higher in the EGFR-TKI-resistant, derivative sub-lines (11–18 GR1-GR6), all of which retained the L858R mutation (Supplementary Table 6). The T790M mutation did not appear in any of the resistant cell lines. The resistant cell lines were morphologically different from the 11–18 parental cells, with four exhibiting a fusiform appearance (Supplementary Fig. 6). No MET amplification was detected in any of the cell lines. Protein and gene expression analyses revealed AXL upregulation in one, upregulation of FGFR1, MET, and Mer in some, and upregulation of EGFR in all of the 11–18 resistant cells (Fig. 5a and Supplementary Fig. 5). HER2 and HER3 levels

were not elevated in the resistant cell lines compared to 11–18 (Supplementary Fig. 5). Low levels of pEGFR were observed in the presence of gefitinib in all cases. Accordingly, significant levels of pERK and pSrc were also observed in most of them (Fig. 5b). Finally, high E-cadherin and beta-catenin expression levels were revealed in the parental cells and were only slightly reduced in the resistant clones. Vimentin expression levels were low in the parental and the resistant cells and no N-cadherin expression was detected in any of the cells (Supplementary Fig. 6a). In conclusion, resistance mechanisms in the 11–18 cells were diverse and differed from those in the PC9 model. No T790M mutations or indication of EMT were detected, and only one cell line showed AXL overexpression, while EGFR was upregulated in all cases. Consequently, while EGFR TKIs completely inactivated the EGFR/ERK pathway in T790M-negative resistant cells derived from PC9, they failed to block activation of ERK and did not completely inhibit EGFR phosphorylation in five of the six resistant cell lines derived from 11–18 cells. Remarkably, increased activation of the Akt pathway was found to be the only common trait between the total of eleven resistant cell lines in these two cell line models.

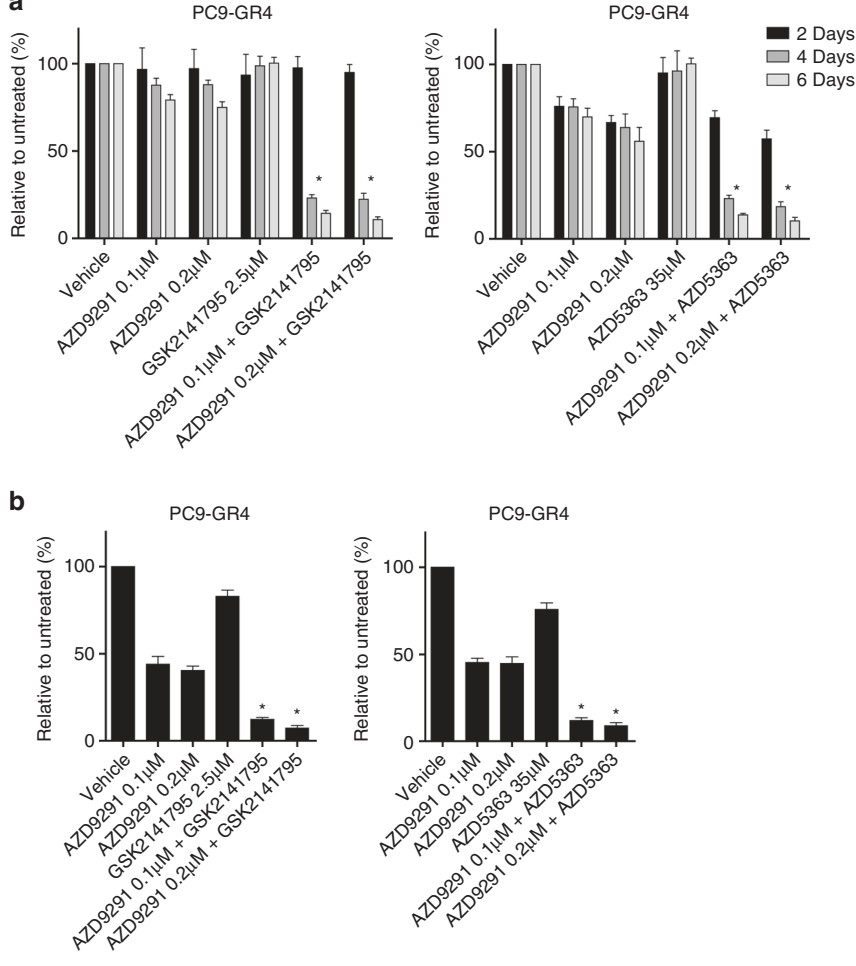

**Fig. 6** Synergistic effect of third-generation EGFR-TKI combined with an Akt inhibitor in T790M-mutated EGFR-TKI-resistant PC9 cells. The combined effects of the third-generation EGFR inhibitor (Osimertinib, AZD9291) targeting the T790M resistance mutation and Akt inhibitors (GSK2141795 or AZD5363) on T790M-mutated gefitinib-resistant PC9-GR4 cells were assessed by **a** CellTiterBlue assay after 2, 4 and 6 days and **b** BrdU incorporation assay performed after 5 days. Concentrations of drugs were chosen to be sub-inhibitory to explore the synergistic potential: AZD9291 0.1 and 0.2 μM; GSK2141795 2.5 μM; and AZD5363 35 μM. All experiments were conducted in 7 replicates (**a**), or quadruplicates (**b**), and results are shown as mean ± SD. *Asterisks* indicate significant difference in ANOVA 1-way test ($p < 0.05$) for the drug combination-treated cells compared to cells treated with either drug alone. For the CellTiterBlue assay significance differences was observed for days 4 and 6

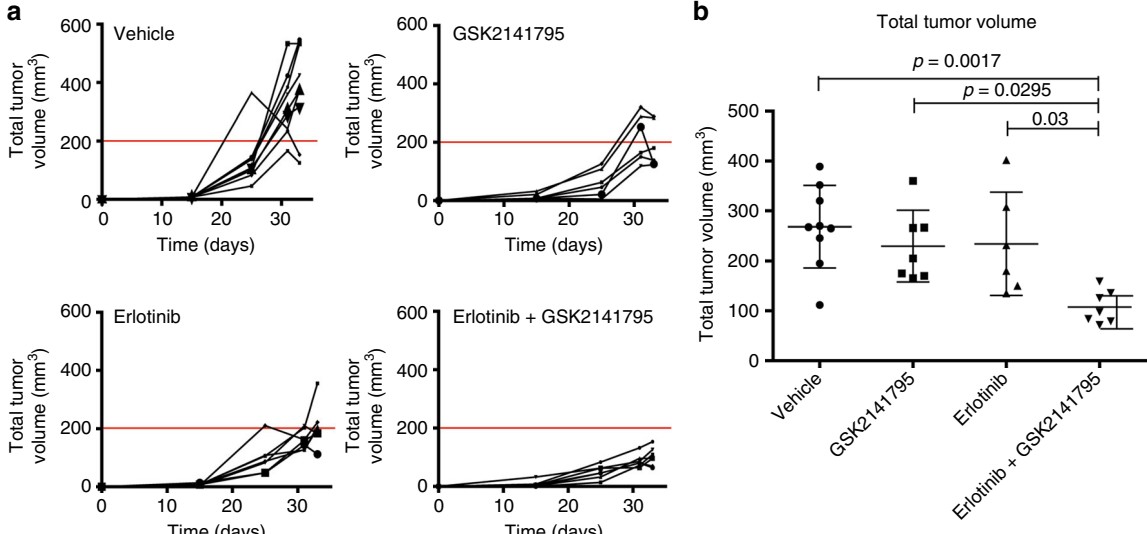

**Fig. 7** Growth co-inhibition of EGFR-TKI-resistant PC9-ER tumors in mice treated with EGFR TKI and Akt inhibitor. **a** Time course assessment of total tumor volume. Each mouse harbored two tumors, and total tumor volume refers to total volume of two tumors per mouse. PC9-ER cells were inoculated on day 0 and treatment was started on day 15. Erlotinib (25 μg/g bodyweight), GSK2141795 (10 μg/g bodyweight) or vehicle was administered by oral gavage once daily for 5 days a week, and mice were killed on day 36. **b** Total tumor volumes measured after excision on day 36. Tumor volumes were calculated by the formula $(length \times weight^2)/2$. Results are shown as mean $\pm$ SD. *Asterisks* indicate significant difference in ANOVA one-way test ($p < 0.05$)

**Synergistic growth inhibition of 11–18 resistant cell lines**. The growth inhibitory effect of the combination of gefitinib with either of the two Akt inhibitors, GSK2141795 and AZD5363, on different 11–18 resistant cell lines was evaluated using CellTiterBlue (Fig. 5c) and BrdU assays (Supplementary Fig. 7). A synergistic growth inhibitory effect was observed in all of the 11–18 EGFR-TKI-resistant cell lines, but not in the parental 11–18 cell line, where the two drugs showed less effect (Fig. 5c and Supplementary Table 4). Similar synergistic growth inhibitory effect on the resistant cell lines was also observed at lower more physiologic concentrations (Supplementary Fig. 8). Western blotting demonstrated a complete inhibition of FOXO1/3a phosphorylation only for the combination of EGFR TKI and Akt inhibitor, whereas PRAS40 phosphorylation was also blocked by the Akt inhibitors as single agent (Fig. 5d). Similarly to what was observed in the PC9 panel, both Akt inhibitors led to an increase in pAKT S473, but still completely prevented phosphorylation of Akt downstream targets, indicating a lack of Akt kinase functional output.

**Third-generation EGFR TKI and Akt inhibitor co-treatment**. To evaluate whether synergistic effect on growth between an EGFR TKI and an Akt inhibitor could also be observed when using a third-generation EGFR-TKI targeting the T790M resistance mutation, we assessed the effects of osimertinib with either of the Akt inhibitors, GSK2141795 and AZD5363. A strong synergistic growth inhibition of the T790M-mutated EGFR-TKI-resistant cell line PC9-GR4 was observed for both combinations as determined by CellTiterBlue and BrdU incorporation assays (Fig. 6). Further, we examined whether the combination of osimertinib and an Akt inhibitor could delay the emergence of resistance in the T790M-positive PC9-GR4 cells compared to the single agents in a colony outgrowth assay. A delay of more than 8 weeks for the combination treatment was observed (Supplementary Fig. 9). Although slightly lower pEGFR levels were present in the resistant compared to parental cells in the absence of gefitinib, EGFR inhibition was critical for growth suppression in these resistant cell lines, as shown by the

synergistic effect of EGFR TKI and Akt inhibitor co-treatment studies (Figs. 3, 5–7 and Supplementary Fig. 9).

**EGFR TKI and Akt inhibitor co-treatment of xenograft mice**. Next, we evaluated the antitumor activity of combined EGFR TKI and Akt inhibitor treatment in a xenograft model. CB17 SCID mice were subcutaneously inoculated in both flanks with $1.5 \times 10^6$ PC9-ER cells that harbor increased pAkt and when palpable, mice were administered erlotinib (25 μg/g bodyweight), GSK2141795 (10 μg/g bodyweight), the drug combination, or vehicle by oral gavage 5 days a week. Weekly measurements using calipers showed extensive growth of the vehicle-treated tumors, a relatively modest antitumor effect of erlotinib or GSK2141795 alone, and a significant inhibitory effect of the drug combination (Fig. 6a). After 3 weeks of treatment, mice were killed and tumors were resected. A significant reduction in tumor volume was observed between the vehicle group and the group receiving combination therapy ($p = 0.0017$), while no significant differences were found between the control group and those treated with erlotinib or GSK2141795 as single agents. Significant differences were also observed when comparing the tumors receiving the combination therapy with those treated with erlotinib ($p = 0.03$) or GSK2141795 ($p = 0.0295$) alone (Fig. 7b and Supplementary Fig. 10a, b).

**pAkt as a biomarker in EGFR-TKI-treated EGFR-mutant patients**. To validate the clinical relevance of our findings, we first evaluated pAKT S473 expression by IHC in baseline samples of 75 EGFR-mutant NSCLC patients treated with first-line EGFR-TKIs (Supplementary Table 7). Importantly, the stability of pAkt staining in IHC analysis was found to be robust independent of fixation time (Supplementary Fig. 11). pAkt expression was determined by a weighted histoscore method also known as the H-score[37], which takes into account the percentage of positive signal and staining intensity. To define the optimal cut-point for pAkt H-score as a continuous variable we applied the method of Contal and O'Quigley[38] that uses log-rank test statistic to estimate the cutpoint[39]. Using this approach 62 was identified as the optimal cut-point divided

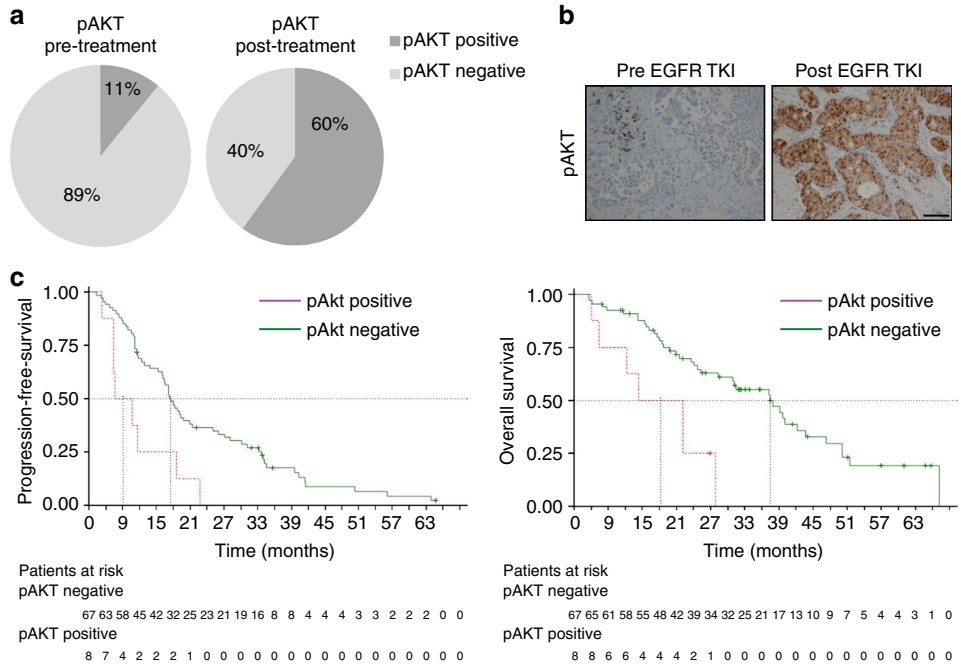

**Fig. 8** Assessment of pAKT in clinical samples. **a** Pie chart (left) shows percentage of pAKT-positive or -negative EGFR-mutant NSCLC patients at baseline treated with first-line EGFR-TKIs. Pie chart (right) shows percentage of pAKT-positive or -negative EGFR-mutant NSCLC patients progressing to first-line EGFR-TKIs. The histoscore value of 62 was used to classify samples as positive (above 62) or negative (below 62) for pAKT. **b** Representative negative and positive immunoshistochemical staining of pAKT. *Scale bar*, 100 μm. **c** *Left*: progression-free survival according to baseline pAKT expression for 75 patients with EGFR-mutant NSCLC treated with first-line EGFR-TKI. Median progression-free survival was 14.5 months (95% CI 12.3–17.9) for the 67 pAKT-negative patients (*green line*) and 6.1 months (95% CI 2.3–15.6) for the 8 pAKT-positive patients (*pink line*); $p = 0.0037$; HR = 0.341; 95% CI 0.159–0.731; $p = 0.0057$. *Right*: OS according to baseline pAKT expression for 75 patients with EGFR-mutant NSCLC treated with first-line EGFR-TKI. Median OS was 34.5 months (95% CI 22.3–39.2) for the 67 pAKT-negative patients (*green line*) and 15.2 months (95% CI 3.0–24.9) for the 8 pAKT-positive patients (*pink line*); $p = 0.0015$; HR = 0.276; 95% CI 0.118–0.646; $p = 0.003$

patients into two groups: pAkt negative (histoscore < 62) and pAkt positive (histoscore ≥ 62). Eight out of 75 (11%) pretreatment samples were pAkt positive (Fig. 8a). PFS to EGFR-TKI treatment was 14.5 months (95% confidence interval (CI), 12.3–17.9) for pAkt negative patients vs. 6.1 months (95% CI, 2.3–15.6) for pAkt-positive patients ($p = 0.0037$), (hazard ratio (HR) 0.341; 95% CI, 0.159–0.731; $p = 0.0057$; Fig. 8b). Overall survival (OS) was 34.5 months (95% CI, 22.3–39.2) for pAkt negative patients vs. 15.2 months (95% CI, 3.0–24.9) for pAkt-positive patients ($p = 0.0015$), (HR 0.276; 95% CI, 0.118 to 0.646; $p = 0.003$; Fig. 8c). A univariate Cox proportional hazards model for PFS and OS was fitted using the following clinically interesting variables: type of EGFR mutation, gender, smoking history, age, brain metastases, and pAkt. Only the status of pAkt contributed significantly to PFS and OS. For a subpopulation of the baseline samples where additional tissue was available we evaluated whether pAkt expression correlated with individual resistance mechanisms by examining the AXL, MET, Her2, and FGFR1 expression and *PIK3CA* mutation status; however, no such correlations were observed (Supplementary Table 8 and Supplementary Fig. 12) and high pAKT levels, although infrequent, were observed across the tumors with each of these known resistance mechanisms. These data reveal a potentially important role for increased pAkt levels as a novel biomarker to predict decreased initial EGFR-TKI response in patients across the spectrum of previously established individual mechanisms of resistance.

We next used IHC to evaluate pAkt in rebiopsy samples from 15 EGFR-mutant NSCLC patients after progression to EGFR TKI. Nine of those 15 (60%) samples were pAkt positive (Fig. 8a and Supplementary Table 9), compared with 11% pAkt

positive in pre-treatment samples. Representative images of pre- and post-treatment samples stained for pAkt are depicted in Fig. 8b. We furthermore evaluated common resistance mechanisms to EGFR TKI in the 15 post-treatment samples, and found that they represented a wide array of well-known resistance mechanisms (Supplementary Table 10), including T790M mutation, PIK3CA mutation, AXL upregulation, MET amplification, or upregulation and alterations in markers of EMT, similarly to what we have observed in our wide variety of our cell line models with acquired EGFR-TKI resistance. Increased pAkt was observed across clinical specimens that have various established resistance-associated alterations (e.g., T790M, high AXL, high MET, and EMT), consistent with our preclinical findings. Importantly, increased pAkt was identified independent of T790M status, indicating that these alterations are neither mutually exclusive nor exclusively coupled in these EGFR-TKI-resistant tumors. Our findings in these clinical samples provide evidence for a previously unappreciated convergent role of Akt activation in acquired EGFR-TKI resistance that is associated with otherwise diverse and unpredictable upstream molecular events.

## Discussion

NSCLC patients with activating EGFR mutations benefit from treatment with EGFR TKIs, but ultimately acquire resistance, which limits PFS to 9–13 months and prevents long-term patient survival[2]. Multiple mechanisms of acquired resistance to EGFR TKI have been described by us and other investigators[10, 12–16]. This diversity, coupled with the unpredictability, of EGFR-TKI resistance mechanisms presents a major challenge for developing

effective treatment regimens to overcome EGFR-TKI resistance. Here, we show for the first time (to our knowledge) that such diversity in acquired resistance mechanisms is associated with convergent activation of the Akt pathway in EGFR-mutant NSCLC. Importantly, we have shown that the combination of an EGFR TKI and an Akt inhibitor elicits synergistic growth inhibition in vitro and significant growth inhibition *in vivo* in otherwise EGFR-TKI-resistant NSCLCs. We propose that by specifically inhibiting Akt together with EGFR, these otherwise diverse resistance mechanisms can be effectively controlled, allowing for a more rational and feasible treatment strategy to address both the molecular heterogeneity of EGFR-TKI resistance[40] and the simultaneous presence of more than one mechanism of resistance (Supplementary Table 8) that we now know characterizes EGFR-TKI progression in many EGFR-mutant NSCLC patients. The therapeutic strategy identified in our study here provides an alternative approach that may more efficiently combat the profound molecular heterogeneity that is an obstacle to the success of current treatments that seek to block an individual upstream resistance mechanism (for instance, an RTK) to overcome acquired EGFR-TKI resistance.

The importance of the Akt pathway in EGFR-TKI resistance was validated in clinical samples from EGFR-mutant NSCLC patients, where phosphorylation of Akt was observed in 60% of tumor samples from patients after progression on EGFR TKIs, but only in 11% of baseline samples. Positivity of pAkt in post-progression samples was found independent of T790M occurrence. Moreover, the presence of pAkt positivity in 8 out of 75 baseline samples correlated with significantly worse PFS and OS to first-line EGFR-TKI treatment (6.1 vs. 14.5 months; $p = 0.0037$ and 15.2 vs. 34.5 months; $p = 0.0015$, respectively). Our clinical data provide new evidence for the potential clinical utility of assessing pAkt levels as a molecular predictor of EGFR-TKI response and resistance in EGFR-mutant NSCLC patients.

Activation of the PI3K-Akt pathway has been reported in EGFR-mutant cell lines and baseline tumors sensitive to TKIs and some reports have associated it with intrinsic sensitivity to first-line treatment with EGFR TKIs[28–32, 41, 42]. However, our report is the first to highlight the importance of Akt activation in cell lines or tumors with acquired resistance to EGFR TKIs and, most importantly, across tumors with a multitude of previously established resistance-promoting mechanisms.

Three EGFR inhibitors, erlotinib and gefitinib, and osimertinib, and two Akt inhibitors, GSK2141795 and AZD5363, were employed in our study to ensure the validity of the observations. We did not attempt knock-down of Akt to verify the effect of the Akt inhibitors, as no single probe can target all three isoforms of Akt simultaneously due to sequence heterogeneity. GSK2141795 and AZD5363 are orally available inhibitors of Akt1-3 with nanomolar or sub-nanomolar potency[35, 43]. Combined EGFR TKI and Akt inhibitor treatment of the EGFR-TKI-resistant NSCLC cells generally induced a complete inhibition of phosphorylation of the two downstream targets PRAS40 and FOXO1/3A. FOXO1/3A is a key pro-apoptotic protein, which, upon phosphorylation by Akt, interacts with 14-3-3 in the nucleus and is transported to the cytoplasm. Phosphorylated FOXO1/3A cannot re-enter the nucleus to trigger transcription of the pro-apoptotic program[23]. PRAS40 functions as a negative regulator of mTORC1, which is a complex involved in protein translation and ribosome biogenesis. Upon phosphorylation by Akt, PRAS40 becomes inhibited and can no longer block activation of the translational machinery and protein synthesis[25]. Interestingly, both Akt inhibitors elicited a feedback hyperphosphorylation of Akt at Ser473, as observed by other investigators[35, 36]. This differs from the effect observed when

employing PI3K or mTOR inhibitors, which result in a decrease in pAkt Ser473[44–46].

Prior studies have evaluated the efficacy of Akt inhibitors in a range of cancers, where promising results had been obtained in preclinical models using drug combinations that include Akt inhibitors, such as AZD5363 combined with fulvestrant in endocrine-resistant breast cancer[47], and AZD5363 combined with AZD8931, an EGFR/ErbB2/ErbB3 inhibitor, in ErbB2-amplified breast cancer[48]. Enhanced tumor growth delay was also observed for GSK2141795 in combination with a MEK inhibitor in models of pancreatic cancer[35]. In the clinic, erlotinib has been combined with the non-ATP competitive pan Akt inhibitor MK-2206 in a phase II clinical trial enrolling advanced NSCLC patients with either mutant or wildtype EGFR[49]. In contrast to our findings reported here, the rationale for this trial was not to block increased Akt activation present in the tumors during EGFR-TKI treatment, as pAkt expression was not examined in these patients, but instead to mitigate hepatocyte growth factor-mediated resistance. In the EGFR-mutant arm of the trial, eligible patients had earlier benefited from, but since progressed on, erlotinib as a single agent. Of 45 EGFR-mutant patients, four had a partial response and 14 had stable disease. Importantly, the level of pAkt was not evaluated in the patients prior to treatment, and we have shown in our study that cell lines with high levels of pAKT are most responsive to combined EGFR and Akt inhibition. However, we have not evaluated resistant cell lines with a low level of pAkt. Based on our new findings, we hypothesize that there would be a correlation between the level of pAKT and the efficacy of the erlotinib and MK-2206 in the patients from the study by Lara and colleagues[49], but this information has not been reported nor have pAKT levels been examined retrospectively through molecular studies in these tumor specimens to our knowledge. Thus, our findings now provide the rationale to do so and pave the way for novel biomarker-driven clinical trials testing an EGFR-TKI plus an Akt inhibitor in appropriately selected EGFR-mutant NSCLC patients with high pAkt tumor expression.

In conclusion, we have shown that Akt pathway activation is commonly associated with acquired resistance to EGFR-TKI treatment in NSCLCs harboring a diverse array of other, previously identified resistance mechanisms. Phosphorylation of Akt was detected in a minority of EGFR-mutant NSCLC patients prior to EGFR-TKI therapy and predicted worse initial EGFR-TKI response. Finally, we have demonstrated that combined treatment with Akt and EGFR inhibitors elicited synergistic growth inhibition in preclinical models of EGFR-TKI resistance. Our findings reveal the unanticipated convergent activation of, and dependence on, Akt in EGFR-TKI resistance that emerges through a multitude of diverse upstream events. While other studies have tested inhibitors of PI3K and/or mTOR in EGFR-mutant NSCLC, our data provide a novel rationale for specifically testing Akt inhibitors in combination with EGFR TKIs in EGFR-mutant NSCLC patients that show Akt activation. We propose that Akt inhibition, specifically, could more uniformly enhance response and survival in patients with high pAkt levels who are at high risk for Akt-mediated resistance, as this distinct approach has the unique potential to combat the otherwise profound heterogeneity of molecular resistance events that are present in EGFR-mutant NSCLC patients with acquired EGFR-TKI resistance to improve their outcomes.

## Methods

**Cell culture and antitumor agents**. Parental PC9 cells were kindly provided by F. Hoffman-La Roche Ltd (Basel, Switzerland) with the authorization of Dr. Mayumi Ono (Kyushu University, Fukuoka, Japan). Parental 11–18 cells were kindly provided by Dr. Mayumi Ono. All cell culture materials were obtained from

Biological Industries (Kibbutz Beit Haemek, Israel) or Gibco Life Technologies (Carlsbad, CA, USA). Cells were cultured in a humidified atmosphere with 5% $CO_2$ at 37 °C in RPMI1640 + 10% fetal bovine serum (FBS), 50 µg/mL penicillin-streptomycin and 2 mM L-Glutamine. EGFR inhibitors (erlotinib, gefitinib, afatinib, dacomitinib, and osimertinib (AZD9291)) and the Akt inhibitor AZD5363 were purchased from Selleck Chemicals (Houston, TX, USA). The Akt inhibitor GSK2141795 (Uprosertib) was purchased from Medchemexpress LLC (Monmouth Junction, NJ, USA). All drugs were dissolved in DMSO, aliquoted and kept at −20 °C.

**Mutation analyses.** DNA was isolated from cell lines and hematoxilin/eosin or IHC slides by standard procedures and genotyped by quantitative PCR with specific probes (Taqman) or standard PCR followed by Sanger sequencing.

**Gene expression analyses.** RNA was isolated from the cell lines in accordance with a proprietary procedure (European patent number EP1945764-B1) as previously described[50]. The primer and probe sets were designed using Primer Express 3.0 Software (Applied Biosystems, Foster City, CA, USA) according to their Ref Seq (http://www.ncbi.nlm.nih.gov/LocusLink). Quantification of gene expression was performed using the ABI Prism 7900HT Sequence Detection System (Applied Biosystems). Expression levels were calculated according to the comparative ΔΔCt method. Commercial RNA controls were used as calibrators (Liver and Lung; Stratagene, La Jolla, CA, USA). For each cell line, three independent experiments were performed.

**Western blotting.** Cells were washed in ice-cold tris buffered saline (TBS), spun down and lysed in radioimmunoprecipitation assay buffer (10 mM Tris HCl, pH 8, 5 mM $Na_2EDTA$, pH 8, 1% NP-40, 0.5% sodium dioxycholate, 0.1% SDS), both containing protease and phosphatase inhibitors (Complete Mini PhosphoSTOP, Roche, Basel, Switzerland). Protein concentrations were determined by Pierce BCA Protein Assay (Thermo Fisher Scientific, Rockford, IL, USA) according to the manufacturer's protocol. 5–40 µg protein was resolved on 4–12% RunBlue SDS-PAGE gels (Expedeon, San Diego, CA, USA), transferred onto PVDF membrane (GE Healthcare Life Sciences, Buckinghamshire, UK), blocked and then incubated with primary monoclonal antibodies ON at 4 °C. Following, the membranes were incubated with goat anti-rabbit, goat anti-mouse (#P0448, #P0447, Dako, Glostrup, Denmark) or donkey anti-goat (#SC-2020, Santa Cruz Biotechnology, Heidelberg, Germany) HRP-conjugated secondary antibodies in 1:5000 dilution for 1 h at room temperature (RT). The immune reactive bands were visualized by Amersham ECL Prime Western Blotting Detecting Reagent (GE Healthcare Life Sciences) and exposed to CL-Xposure film (Thermo Fisher Scientific).

**Immunohistochemistry.** IHC was performed on 5 µm sections using an auto-mated immunostainer (Ventana BenchMark ULTRA, Ventana Medical Systems, Oro Valley, AZ, USA) and protein expression was quantified using the histoscore method as previously described[51]. The following antibodies were used: AXL (Cell Signaling Technology, Leiden, The Netherlands, #8661, dilution 1:100), pAkt (Cell Signaling Technology, #4060, dilution 1:50), phospho-PRAS40 (Cell Signaling Technology, #2997, dilution 1:100), E-cadherin (Roche #5973872001, RTU), β-catenin (Roche #5269016001, RTU), vimentin (Roche #5278139001, RTU), and N-cadherin (Abcam, Cambridge, UK, #ab18203, dilution 1:100).

**Cell growth and apoptosis assays.** Cells were seeded at 4000 cells per well in 96-well plates, allowed to attach for 24 h and treated for 72 h with drug. After treat-ment, cells were incubated with medium containing MTT (0.75 mg/mL in med-ium) for 1 h at 37 °C. Culture medium with MTT was then removed and formazan crystals dissolved in 100 µL DMSO (Sigma-Aldrich, St. Louis, MO, USA). Cell numbers were estimated by measuring the absorbance at 495 nm using a micro-plate reader (BioWhittaker, Walkersville, MD, USA).

For drug combination experiments, 10,000 cells per well were seeded in 24-well plates and left to attach for 6 h before drugs or vehicle were added; then grown at 37 °C. Cell viability was quantified by crystal violet staining or by CellTiterBlue (Promega, WI, USA) and cell proliferation was evaluated by BrdU incorporation using the BrdU Cell Proliferation Assay Kit (Cell Signaling Technology, Beverly, MA, USA), according to the manufacturer´s instructions. Crystal violet assay was performed by adding staining solution for 5 min at RT, washing cells twice in $H_2O$, redissolving in Na-citrate buffer (29.41 g Na-citrate in 50% EtOH) and measuring the absorbance at 570 nm. Apoptosis was assessed using the Cell Death Detection ELISA$^{Plus}$ kit (Roche, Basel, Switzerland) according to the manufacturer's instructions. Colony outgrowth assay was performed as described in Tricker et al. (2015)[52].

**SILAC labeling.** Cells were propagated in RPMI1640 (BioNordika, Herlev, Denmark) supplemented with 580 mg/L-glutamine and 200 mg/L proline and 10% dialyzed FBS (Gibco, Life Technologies, Carlsbad, CA, USA). PC9-ER cells were propagated in "heavy" media containing 75 mg/L $^{13}C_6$-Lys and 28 mg/L $^{13}C_6$-Arg

(Cambridge Isotope Laboratories, Tewksbury, MA, USA), and PC9 cells were propagated in "light" medium containing 75 mg/L $^{12}C_6$-Lys and 28 mg/L $^{12}C_6$-Arg (Sigma-Aldrich). Cells were propagated for six passages in SILAC media and harvested to ascertain complete isotope incorporation.

**Cell harvest for mass spectrometry.** All cells were harvested at 80% confluence. Cells were washed twice with ice-cold TBS and lyzed with ice-cold 0.1 M $Na_2CO_3$ pH 11 lysis buffer containing 1 mM activated sodium pervanadate and protease (Complete Mini EDTA-free, Roche, Basel, Switzerland) and phosphatase (Complete Mini PhosSTOP, Roche, Basel, Switzerland) inhibitors. Cell scrapers were used not to damage extracellular membrane proteins. The lysate was adjusted to 1 mM $MgCl_2$ and 3 µL benzonase (Sigma-Aldrich) was added and samples were left on ice for 15 min to degrade RNA and DNA. Lysates were now homogenized using a Branson sonifier 250, 2 × 30 s, output 10, output control 2.5. The lysates were then centrifuged at 100,000×g for 45 min at 4 °C in a Sorvall RC M150 GX ultracentrifuge to separate soluble proteins (supernatant) from membraneous proteins (pellet). The pellets were washed with 0.5 M triethylammonium bicarbonate (TEAB) followed by 0.05 M TEAB to remove soluble protein contamination.

**Protein purification and digestion.** The supernatant proteins were precipitated by adding four volumes of ice cold acetone and left at −20 °C followed by centrifugation at 7000×g for 15 min. The pellets containing soluble proteins were then dissolved in 8 M urea and incubated to fully dissolve protein clusters. Protein concentrations were determined by the Bio-Rad Protein Assay before proteins were reduced by 20 mM dithiothreitol for 45 min and subsequently alkylated by 40 mM iodoacetamide for 45 min in the dark. Samples were then digested with Lys-C (Sigma-Aldrich) 1 µg enzyme/50 µg protein for 5 h at RT. Samples were diluted eight times with 0.1 M ammonium bicarbonate and digested with trypsin (1 µg enzyme/100 µg peptide) at 37 °C. Samples were acidified to 0.1% trifluoroacetic acid (TFA) and centrifuged at 15,000×g for 10 min to pellet insoluble materials such as lipids, and the supernatant was kept for further analysis. The membrane proteins were dissolved in 8 M urea, reduced and alkylated as stated above before 0.5 µL Sialidase A (Europa Bioproducts, Cambridge, UK) and 1 µL PNGase F (Sigma-Aldrich) was added at 37 °C to remove extracellular glycan structures. Then, samples were treated with Lys-C and trypsin as described above. Digested peptides were desalted on in-house packed stage tip columns composed of two C18 membrane disks (Empore 3 M, Bellefonte, USA) and porous R2/R3 reverse-phase resins (Applied Biosystems). In brief, samples were acidified to pH ~2 before peptides were applied to 0.1% TFA pre-equilibrated columns, washed with 0.1% TFA and eluted using 70% ACN, 0.1% TFA.

**HILIC fractionation.** Samples were adjusted to ~40 µL of 90% ACN, 0.1% TFA by first dissolving it in 0.4 µL of 10% TFA, then adding 3.6 µL of $H_2O$, and finally adding 36 µL of ACN. The samples were injected onto an in-house packed TSKgel Amide-80 HILIC 320 µm × 170 mm capillary HPLC column using an Agilent 1200 capillary HPLC system. The peptides were eluted using a gradient from 90% ACN, 0.1% TFA to 60% ACN, 0.1% TFA for over 46 min at a flow rate of 6 µL/min. The fractions were automatically collected in a microwell plate at 1 min intervals after UV detection at 210 nm, and the fractions were pooled according to the UV detection to a total of 10 fractions. The fractions were dried by vacuum centrifugation. Prior to LC–MS/MS the samples were redissolved in 0.5 µL of 100% formic acid (FA) and diluted with $H_2O$ to 5.5 µL. A total of 5 µL of each fraction were analyzed by reverse-phase nanoLC–MS/MS.

**NanoLC-MS/MS and data analysis.** The peptides were loaded onto an Easy-nanoLC (Thermo Fisher Scientific) coupled to an Orbitrap FusionTM TribridTM mass spectrometer (Thermo Fisher Scientific). Peptides were loaded onto a pre-column (2 cm Reprosil—Pur C18 AQ 5 µm RP material (Dr. Maisch, Ammerbuch-Entrigen, Germany)) using the EASY-LC system and eluted directly onto a 20 cm-long fused silica capillary column (75 µm ID) packed with Reprosil—Pur C18 AQ 3 µm RP material. The peptides were separated using a gradient from 0–34% B (A buffer: 0.1 % FA; B buffer: 90% ACN/0.1% FA) at a flow rate of 250 nL/min over 30–90 min depending on the UV trace of the HILIC fractions. Peptides ($m/z$ 400–1400) were analyzed in full MS mode using a resolution of 120.000 FWHM at 200 m/z and the peptides were selected and fragmented using high-energy collisional dissociation (HCD) and the fragment ions were recorded in the LTQ with low resolution (rapid scan rate). A maximum of 3 s were allowed between each MS and for MSMS the ion filling time was set to 35 ms and an AGC target value of 2E4 ions. The mass spectrometry data were processed using Proteome Discoverer software v1.4 (Thermo Scientific), and the raw data were searched in the Swissprot and Uniprot databases using the Mascot and SEQUEST search algorithms. Quantitation using SILAC was performed in Proteome Discoverer v 1.4 using the SILAC Quantitation node.

We calculated the average of the triplicate ratios for each protein and selected those that fulfilled two criteria. First, the average ratio should be ≥2-fold differentially expressed, and second, at least two of the triplicate ratios should be ≥2-fold. We then eliminated the cytoplasmatic and membrane proteins that were only up or downregulated in the membrane or cytoplasmic fraction, respectively.

Ingenuity Pathway Analysis software (Qiagen, Redwood City, CA, USA) was used for classifying proteins in functional groups.

**Mice xenograft study**. All animal experiments were approved by The Experimental Animal Committee of The Danish Ministry of Justice and were performed at the animal core facility at University of Southern Denmark. Mice were housed under pathogen-free conditions with ad libitum food and water. Subconfluent PC9-ER cells (1.5 x 10$^6$) were harvested using accutase and resuspended in a 1:1 mixture of extracellular matrix from Engelbreth-Holm-Swarm sarcoma (Sigma-Aldrich) and RPMI1640 media, and injected subcutaneously into 16-week-old female CB17 SCID mice (Taconic, Ejby, Denmark). When tumors reached a palpable diameter of 2–3 mm, mice were randomized into groups of $n =$ 9 per group. Animals were euthanized if they showed any adverse signs of disease including weight loss, paralysis, thymus dysfunction, or general discomfort. Accordingly, eight mice were censored during the course of the study. Erlotinib HCl was formulated at 25 μg/g bodyweight in 15% Captisol (La Jolla, CA, USA). GSK2141795 was formulated at 10 μg/g bodyweight in DMSO. 15% Captisol was used as vehicle, and the concentration of DMSO did not exceed 10% when administered. Drugs were administered 5 days a week for 3 weeks by oral gavage. Maximum volume per mouse was 200 μL. Tumor volume and bodyweight were surveyed during the extent of the study. Tumor volume was measured with calipers and calculated according to: tumor volume (mm$^3$) = (length×width$^2$)/2.

**Patient samples collection**. Clinical data were assessed and tumor samples studied in accordance with an approved protocol of the institutional review board of Germans Trias i Pujol Hospital, Badalona, Spain and de-identified for patient confidentiality.

**Statistical methods**. PFS and OS were estimated by the Kaplan–Meier method, and the nonparametric log-rank test was used to evaluate differences between groups. Cox proportional hazard regression model was applied with pAKT as covariate, obtaining HR and their 95% CIs. Each analysis was performed with the use of a two-sided 5% significance level and a 95% CI. Statistical analyses were performed using SAS version 9.3. Synergy was defined as the two drugs together having an effect greater than the sum of the two drugs separate effects (CI<1).

**Data availability**. All relevant data are available from the authors upon request.

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

## Acknowledgements

We thank the following investigators for providing us with clinical tumor samples: Andres Felipe Cardona, Clinical and Translational Oncology Group-FICMAC, Colombia; Guillermo Lopez Vivanco, Hospital de Cruces de Barakaldo, Bizcaia, Spain; Alain Vergnenegre, Service de Pathologie Respiratoire et d'Allergologie; Jose Miguel Sanchez, Hospital La Princesa, Madrid, Spain; Mariano Provencio, Hospital Puerta de Hierro, Madrid, Spain; Filippo de Marinis, Divisione di Oncologica Toracica, Istituto Europeo di Oncologia, Milano, Italy; Enric Carcereny, Institut Català d'Oncologia, Hospital Germans Trias i Pujol, Badalona, Spain; Noemi Reguart, Hospital Clínic, Barcelona, Spain; Charo Garcia Campelo, Hospital A Coruña, A Coruña, Spain; Eric Santoni-Rugiu, Rigshospitalet, Copenhagen, Denmark. Finally, we thank M.K. Occhipinti for editorial assistance. The work in Dr. Ditzel's laboratory was supported by the Danish Cancer Society, and Sino-Danish Centre for Education and Research, National Experimental Therapy Partnership (NEXT) Bioinformatics financed by Innovation Fund Denmark, and Academy of Geriatric Cancer Research (AgeCare). The work in Dr. Rosell's laboratory was supported by grants from the La Caixa Foundation and Red Tematica de Investigacion Cooperativa en Cancer (RTICC; grant RD12/0036/ 0072). Dr. Larsen was supported by the Lundbeck Foundation (Junior Group Leader Fellowship) and a grant from the VILLUM Foundation to the VILLUM Center for Bioanalytical Sciences. Dr. Bivona was supported by grants from NIH/NCI R01CA169338 and Pew and Stewart Foundations. We also thank the animal core facility at University of Southern Denmark for maintenance and care of the mice in this study.

## Author contributions

K.J., M.A.M., H.J.D. and R.R. designed the study. K.J., J.B.-A., A.G.-C., M.G., C.C.-S., J.C.-S., M.A.M. and M.R.L. contributed with in vitro data. K.J. and M.H.P. performed the animal study. N.K. and C.T. contributed with data on clinical cohorts. U.L., E.F. and S.V. contributed with clinical samples. A.D. contributed with statistical analysis. K.J., N.K., M.A.M. and H.J.D. wrote the manuscript. T.G.B. revised the manuscript. R.R., M.A.M. and H.J.D. supervised the project and revised the manuscript.

## Additional information

**Competing interests:** The authors declare no competing financial interests.

