## [Peer Review File · Nature Communications]

Reviewers' comments:

Reviewer #1 (Remarks to the Author):

The authors have presented data demonstrating activation of AKT as a common resistance mechanism in EGFR mutation positive, EGFR TKI resistant NSCLC that occurs regardless of the upstream resistance mechanism.

1. Although there is an inference that AXL, MET, MER and other RTK up-regulation drive pAKT in EGFR TKI resistant cell lines, the authors have not demonstrated this by either drug inhibition or knockdown studies.

2. The experiments do not demonstrate how pAKT is upregulated in T790M+ resistant lines. This is important as it suggests concurrent or prior alterations that regulate this independently of EGFR.

3. It is interesting to note that pEGFR is decreased in the absence of EGFR inhibition in the 11-18 GR4 cells. Was this the case with other GR lines derived from 11-18 and what is the reason for the decreased pEGFR as these cells still harbor EGFR L858R which is constitutively activated?

4. The 75 NSCLC tumor samples should be tested for PIK3CA mutations as this has been previously described as impacting PFS on EGFR TKI for EGFR mutant lung cancers.

5. It is disappointing that the clinical trial of an AKT inhibitor only showed a ~10% response rate because the post-EGFR TKI samples in this study would have predicted that a large portion of these 45 patients (even if not quite 60%) would be pAKT positive and thus might temper enthusiasm for a clinical trial, even if they did not perform pAKT testing.

Other minor points that should be addressed include the following:

1. In the introduction, this sentence does not appear to make sense as both clauses say the same thing: Here, we show that Akt pathway activation is not only observed as an acquired resistance mechanism following EGFR TKI treatment, but furthermore constitutes a common trait in EGFR-mutant NSCLCs with acquired resistance to EGFR TKIs. (p6 line 136)

Reviewer #2 (Remarks to the Author):

In this manuscript Jacobsen et al explore the role of Akt as a marker and therapeutic target of resistance to EGFR TKI therapy. They demonstrate in 2 different cell line models that despite the heterogeneity of resistance to EGFR TKI therapy in vitro, that they almost universally converge on AKT activation. They further identify synergy with two different AKT inhibitors in established resistant cell lines, which coincided with effective blockade of PRAS40 inhibition as a downstream readout, in vitro and in vivo using a mouse xenograft model. Finally they demonstrate that elevated pAKT levels co-occur with acquired resistance to EGFR TKIs in patients, and that high pAKT levels are a poor prognostic sign.

In general, this is a well conducted study. Since the concept of inhibiting AKT is not entirely novel, despite several unique aspects presented here, additional work is necessary to rise to the level of Nature Communications, specifically about the comparative role of MEKi vs AKTi at preventing the establishment of resistance, since MEK/ERK inhibition has been shown to be at least as effective in this setting, and it would be important to compare these 2 different combinations as it relates to a clinical development strategy with 3rd generation EGFR TKIs.

Specifically,

1. How does single agent AZD9291, MEKi, AKTi, or combination AZD9291/MEKi vs AZD9291/AKTi

fare at preventing resistance in PC9 +/- 11-18 colony outgrowth assays (see Tricker et al., Cancer Discov 2015; PMID 26036643)

2. This is especially relevant since in Figs 4 and 5 erlotinib + AKTi inhibits pERK in parental PC9 and 11-18 but not resistant clones. The authors should at a minimum discuss the potential consequences of unchecked pERK activation in this setting, especially as the in vivo PDX treatment response is incomplete, and the other reason for unimpressive clinical trial results in the advanced setting may be due to unchecked MAPK pathway activation.

3. Concentrations of erlotinib (30 microM) as well as AZD5363 (35 microM) are not physiologic and likely associated with off-target effects. The authors should repeat at least a subset of the experiments in established resistant cell lines using 1 microM erlotinib and physiologic concentrations of AKTi to explore synergy.

Below please find a point-by-point response to the reviewers' comments.
Revisions in the manuscript are highlighted.

Reviewer #1:

The authors have presented data demonstrating activation of AKT as a common resistance mechanism in EGFR mutation positive, EGFR TKI resistant NSCLC that occurs regardless of the upstream resistance mechanism.

1. Although there is an inference that AXL, MET, MER and other RTK up-regulation drive pAKT in EGFR TKI resistant cell lines, the authors have not demonstrated this by either drug inhibition or knockdown studies.

Our Response: As requested by the reviewer we have performed a series of drug inhibition experiments to clarify this point. We have tested the FGFR inhibitor nintedanib in the PC9-GR5 cell line, overexpressing FGFR1; and we have observed a dose-dependent inhibition of pAkt in presence of the drug. We have also observed a decrease in pAkt levels in the PC9-GR2 cell line (with MET activation) in presence of crizotinib. The AXL inhibitor BGB324 only had a minor effect on pAkt in the PC9-ER cell line, overexpressing AXL. In each case, however, suppression of p-AKT levels remains incomplete with upstream RTK inhibition likely because multiple RTKs (or additional events) can cooperate to promote AKT signaling in these cells. Taken together, these results support the idea that, at least in some cases, MET and FGFR1 overexpression or activation promotes p-Akt in EGFR TKI resistant cell lines but that multiple events can contribute to AKT activation as well. We have included this data as a new Supplementary Figure 4 as well as in Results (p 10).

These findings underscore the relevance of our discovery that p-AKT is a convergent resistance node. One does not need to know precisely which upstream RTK or combination of RTKs, or other mechanism is driving p-AKT activation (in some cases perhaps cooperatively) to promote resistance – inhibition of this convergently-activated AKT overcomes EGFR TKI resistance in a manner that is agnostic to the underlying molecular event(s) driving AKT activation. Thus, our data provide a potential therapeutic solution – via AKT inhibitor treatment – to the profound heterogeneity underlying EGFR TKI resistance.

2. the experiments do not demonstrate how pAKT is upregulated in T790M+ resistant lines. This is important as it suggest concurrent or prior alterations that regulate this independently of EGFR.

Our analyses have shown that AZD9291 induces a moderate decrease of pAkt in T790M+ PC9-GR1 cells. These cells also show MET activation and we have also observed that crizotinib moderately blocks AKT phosphorylation in PC9-GR1. Thus, the pAKT is most likely a consequence of receptor kinase co-activation (i.e. MET, AXL, ERBB2, etc.) together with EGFR activation elicited by the T790M mutation.. Another possibility would be that there is a mutation or copy number alteration in PTEN/PI3K/AKT/mTOR signaling genes in the cells – however our DNA sequencing and FISH have ruled out mutations/copy number alterations in PIK3CA. mTOR was not tested. We have included this data as a new Supplementary Figure 4d as well as in Methods (p 22) and Results (p 10).

3. It is interesting to note that pEGFR is decreased in the absence of EGFR inhibition in the 11-18 GR4 cells. Was this the case with other GR lines derived from 11-18 and what is the reason for the decreased pEGFR as these cells still harbor EGFR L858R which is constitutively activated?

Our Response: We agree with the reviewer that it appears as pEGFR is decreased in 11-18 GR4 even in the absence of EGFR inhibitor compared to the parental 11-18 cell line. We have repeated the experiment and it seems clear that a solid pEGFR band for 11-18 GR4 is observed in the absence of EGFR inhibitor, however when the data for 11-18 and 11-18 GR4 are shown on the same blot, the pEGFR in the absence of EGFR inhibitor is slightly higher in 11-18 compared to 11-18 GR4. To avoid confusing the readers we have replaced the pEGFR blot in Fig.5d with one slightly more exposed where nice pEGFR bands are shown for 11-18 GR4, while the pEGFR for 11-18 is slightly overexposed. We have also now added a sentence to Results (p 14) explaining that slightly lower pEGFR levels were observed in resistant vs parental cells in the absence of gefitinib. Similar slightly lower pEGFR was also observed for the other 11-18 GR cell lines (11-18 GR1, 11-18 GR2 and 11-18 GR3).

4. The 75 NSCLC tumor samples should be tested for PIK3CA mutations as this has been previously described as impacting PFS on EGFR TKI for EGFR mutant lung cancers.

Our Response: As requested by the reviewer we have evaluated the PIK3CA mutation status in 63 of the 75 NSCLC tumor samples where remaining tissue was available. PIK3CA mutations were only identified in 3 samples, one positive and two negative for pAkt staining. Of 7 pAkt+ samples, only one was positive for PIK3CA mutations, indicating that pAkt is not driven by PIK3CA mutations in a majority of patients in this setting. We have included the data in supplemental Table 8 as well as in Methods (p 22) and Results (p 16).

5. It is disappointing that that the clinical trial of an AKT inhibitor only showed a ~10% response rate because the post-EGFR TKI samples in this study would have predicted that a large portion of these 45 patients (even if not quite 60%) would be pAKT positive and thus might temper enthusiasm for a clinical trial, even if they did not perform pAKT testing.

Our Response: This is an important point and we thank the Reviewer for raising it. We agree with the Reviewer that our data would predict a higher response rate for AKT inhibition combined with an EGFR inhibitor in EGFR mutant patients. However, an important novel aspect of our manuscript is our data suggesting an important predictive role of increased pAKT expression as a biomarker of response to AKT inhibition (together with an EGFR TKI) in EGFR mutant patients. As the Reviewer correctly points out, pAKT testing was not performed in the clinical specimens from patients enrolled on the prior clinical trial. Given this lack of biomarker data, we believe that firm conclusions regarding the potential efficacy of AKT inhibitor treatment in patients with acquired EGFR TKI resistance cannot be drawn from that study. We believe our data pave the way for a biomarker-driven re-evaluation of the clinical efficacy of AKT inhibitor treatment, combined with an EGFR TKI, specifically in patients whose tumors harbor increased pAKT levels in prospectively studied clinical cohorts; we are enthusiastic about pursuing this question clinically as a next step, following publication of our manuscript.

Other minor points that should be addressed include the following:

1. In the introduction, this sentence does not appear to make sense as both clauses say the same thing:

Here, we show that Akt pathway activation is not only observed as an acquired resistance mechanism following EGFR TKI treatment, but furthermore constitutes a common trait in EGFR-mutant NSCLCs with acquired resistance to EGFR TKIs. (p6 line 136)

Our Response: As requested by the reviewer we have reworded the sentence in Introduction (page 6).

Reviewer #2:

In this manuscript Jacobsen et al explore the role of Akt as a marker and therapeutic target of resistance to EGFR TKI therapy. They demonstrate in 2 different cell line models that despite the heterogeneity of resistance to EGFR TKI therapy in vitro, that they almost universally converge on AKT activation. They further identify synergy with two different AKT inhibitors in established resistant cell lines, which coincided with effective blockade of PRAS40 inhibition as a downstream readout, in vitro and in vivo using a mouse xenograft model. Finally they demonstrate that elevated pAKT levels co-occur with acquired resistance to EGFR TKIs in patients, and that high pAKT levels are a poor prognostic sign.

In general, this is a well conducted study. Since the concept of inhibiting AKT is not entirely novel, despite several unique aspects presented here, additional work is necessary to rise to the level of Nature Communications, specifically about the comparative role of MEKi vs AKTi at preventing the establishment of resistance, since MEK/ERK inhibition has been shown to be at least as effective in this setting, and it would be important to compare these 2 different combinations as it relates to a clinical development strategy with 3rd generation EGFR TKIs.

Specifically,

1. How does single agent AZD9291, MEKi, AKTi, or combination AZD9291/MEKi vs AZD9291/AKTi fare at preventing resistance in PC9 +/- 11-18 colony outgrowth assays (see Tricker et al., Cancer Discov 2015; PMID 26036643)

Our Response: As proposed by the reviewer we have performed colony outgrowth assays using the protocol setup as described by Tricker et al. A more than 8-week delay in the growth of PC9 GR4 using the combination treatment of AZD9291 and uprosertib compared to single AZD9291 was observed. This delay is in the range of that reported for osimertinib plus MEKi in the article from Tricker et al. The results have been included in Supplementary Figure 9 and commented in Results (p 14).

2. This is especially relevant since in Figs 4 and 5 erlotinib + AKTi inhibits pERK in parental PC9 and 11-18 but not resistant clones. The authors should at a minimum discuss the potential consequences of unchecked pERK activation in this setting, especially as the in vivo PDX treatment response is incomplete, and the other reason for unimpressive clinical trial results in the advanced setting may be due to unchecked MAPK pathway activation.

Our Response: This is an important point that implies that perhaps higher order combinations of agents might be required for complete tumor regressions in this setting. One example is combined EGFR, AKT and MEK or ERK inhibitor treatment. This triple therapy regimen would however be predicted to show substantial clinical toxicity. Thus, while it is possible to test this therapeutic approach in mouse models, the clinical applicability is uncertain. We intend to explore this question in a separate series of dedicated studies in cell lines and mouse models.

3. Concentrations of erlotinib (30 microM) as well as AZD5363 (35 microM) are not physiologic and likely associated with off-target effects. The authors should repeat at least a subset of the experiments in established resistant cell lines using 1 microM erlotinib and physiologic concentrations of AKTi to explore synergy.

Our Response: As requested by the reviewer we examined the synergistic effect of EGFR-inhibitors and AZD5363 at more physiologic concentrations in two of the resistant cell lines, 11-18 GR4 and PC9 GR2. Nice synergic growth inhibition with 1, 2 and 5 microM gefitinib and 5-10 microM AZD5363 was

observed while the single agents caused no growth inhibition at these concentrations. The data are presented as a Supplemental Figure 8 and described in Results (p 13). Further, the animal model likely more closely reflects the “real patient” and here synergism was observed when administering mice with drug doses that lead to physiologic concentrations in blood.

We hope these revisions adequately address the comments of the reviewers and render the manuscript acceptable for publication.

Reviewers' comments:

Reviewer #1 (Remarks to the Author):

This is a well-written manuscript that describes mechanisms of resistance to EGFR TKI inhibitors in EGFR mutant, NSCLC. The authors demonstrate that AKT signaling is a common activated pathway, regardless of the upstream driving pathway (MET, AXL, FGFR, etc.). This has important potential clinical implications as it may be challenging to develop personalized drug combinations of EGFR + AXL or EGFR + FGFR. The authors combine in vitro and in situ data, which support their hypothesis that AKT is a common convergent resistance pathway for EGFR TKI resistance in both EGFR TKI naive and EGFR T790M+ cancer.

Criticisms:

1. If PIK3CA/AKT/MTOR critical for EGFR response, why did concurrent PIK3CA mutations in EGFR mutation positive tumors not impact response rate or time to progression in patients treated with EGFR TKIs (Eng et al., JTO 2016)?
2. The authors should demonstrate how MET activated in PC-9 GR1 and GR2 as MET levels unaltered compared to parental line?
3. It is not clear whether EGFR inhibition is still needed at resistance in the PC-9 or 11-18 resistant cell lines as for example pEGFR is down in PC9-ER in the absence of EGFR inhibition. Resistance can often be accompanied by a true oncogene switch where the original primary oncogene is no longer activated. pEGFR in the absence of EGFR TKI should be shown in the resistant cell lines (and in the resistant tumor samples). This has implications for how you would design a clinical trial because EGFR + AKT inhibition may lead to greater toxicity.
4. Related to above, PC-9GR1, despite not having a T790M mutation has poor inhibition of pEGFR with EGFR TKI suggesting maintenance of EGFR signaling in this cell line. Can you describe the mechanism.
5. The response rate with EGFR + AKT inhibitor referenced in the discussion was disappointing with less than a 10% response rate in the 45 patients with EGFR mutant, EGFR TKI resistant NSCLC. Given the authors data on increased pAKT in resistant samples of 60% (Figure 8), this would suggest while AKT is important, inhibiting AKT is not sufficient to overcome resistance in the majority of patients.

Reviewer #2 (Remarks to the Author):

The authors have satisfactorily addressed my concerns

We appreciate that Reviewer 2 finds that we have satisfactorily responded to all the concerns raised. We also appreciate the insightful additional feedback of reviewer 1. Below please find our point-by-point response. Revisions in the manuscript are highlighted.

Reviewer #1:

1. If PIK3CA/AKT/MTOR critical for EGFR response, why did concurrent PIK3CA mutations in EGFR mutation positive tumors not impact response rate or time to progression in patients treated with EGFR TKIs (Eng et al., JTO 2016)?

Our Response: Only 6 patients harboring both EGFR and PIK3CA mutations were included in the study by Eng et al., 2015, PMID:26334752, thus it is difficult to make firm conclusions from this study.

2. The authors should demonstrate how MET activated in PC-9 GR1 and GR2 as MET levels unaltered compared to parental line?

Our Response: This could be caused by multiple mechanisms, such as via HGF secretion, possible presence of splicing variants or mutations of MET modifying the kinase domain. However, we respectfully feel this question while interesting is beyond the scope of this study and the results will not affect the overall conclusion of the study.

3. It is not clear whether EGFR inhibition is still needed at resistance in the PC-9 or 11-18 resistant cell lines as for example pEGFR is down in PC9-ER in the absence of EGFR inhibition. Resistance can often be accompanied by a true oncogene switch where the original primary oncogene is no longer activated. pEGFR in the absence of EGFR TKI should be shown in the resistant cell lines (and in the resistant tumor samples). This has implications for how you would design a clinical trial because EGFR + AKT inhibition may lead to greater toxicity.

Our Response: We fully agree with the reviewer that determining whether EGFR inhibition is still required for inhibition of the resistant PC-9 or 11-18 cell lines is important as it may influence the design of clinical trials with AKT inhibitors in this setting. However, first, it should be emphasized that although the pEGFR level is lower in the resistant (11-18 G1-GR4) vs. parental 11-18 cells, substantial pEGFR levels still remain in the resistant cells. Secondly, outgrowth and growth inhibition experiments as well as mouse studies clearly demonstrate synergism between Akt inhibitors and EGFR TKIs (Figs. 3, 5, 6 and 7 and Supplementary Fig.9) demonstrating that EGFR inhibition is needed. We have added this information to the Results (page 14).

4. Related to above, PC-9GR1, despite not having a T790M mutation has poor inhibition of pEGFR with EGFR TKI suggesting maintenance of EGFR signaling in this cell line. Can you describe the mechanism.

Our Response: We believe the reviewer may have misread Suppl. Table 3 as PC9-GR1 actually has the T790M mutation as indicated in the table.

5. The response rate with EGFR + AKT inhibitor referenced in the discussion was disappointing with less than a 10% response rate in the 45 patients with EGFR mutant, EGFR TKI resistant NSCLC. Given the authors data on increased pAKT in resistant samples of 60% (Figure 8), this would suggest while AKT is important, inhibiting AKT is not sufficient to overcome resistance in the majority of patients.

Our Response: As already discussed in the Discussion (page 20), the rationale for the above mentioned trial was not to block increased Akt activation present in the tumors during EGFR TKI treatment, as pAkt expression was not examined in these patients, but instead to mitigate hepatocyte growth factor- (HGF) mediated resistance. As the reviewer mentioned, four of the 45 EGFR-mutant patients had a partial response while 14 had stable disease. Importantly, the level of pAkt was not evaluated in the patients prior to treatment, and we have shown in our study that cell lines with high levels of pAKT are most responsive to combined EGFR and Akt inhibition. Finally, the Akt inhibitor used in a phase II clinical trial was a non-ATP competitive pan Akt inhibitor (MK-2206), whose mode of action may differ from AZD5363 and GSK2141795 used in our study.

We hope these revisions adequately address the comments of the reviewers and render the manuscript acceptable for publication.

REVIEWERS' COMMENTS:

Reviewer #1 (Remarks to the Author):

The authors have satisfactorily addressed the concerns and criticisms.